# Assessing species composition and insecticide resistance of *Anopheles gambiae* complex members in three coastal health districts of Côte d'Ivoire

Jackson K. I. Kouamé[1,2]*, Constant V. A. Edi[2], Julien B. Z. Zahouli[2,3], Ruth M. A. Kouamé[2,4], Yves A. K. Kacou[1,2], Firmain N. Yokoly[1,2], Constant G. N. Gbalegba[5], David Malone[6], Benjamin G. Koudou[1,2]

1 Unité de Formation et de Recherche Sciences de la Nature, Université Nangui Abrogoua, Abidjan, Côte d'Ivoire, 2 Environnement et Santé, Centre Suisse de Recherches Scientifiques en Côte d'Ivoire, Abidjan, Côte d'Ivoire, 3 Centre d'Entomologie Médicale et Vétérinaire, Université Alassane Ouattara, Bouaké, Côte d'Ivoire, 4 École Supérieure d'Agronomie, Institut National Polytechnique Félix Houphouët Boigny, Yamoussoukro, Côte d'Ivoire, 5 Unité de Lutte Antivectorielle, Programme National de Lutte Contre le Paludisme, Abidjan, Côte d'Ivoire, 6 Innovative Vector Control Consortium, Liverpool School of Tropical Medicine, Liverpool, United Kingdom

* jackson.kouame@csrs.ci, jacksonkouame2017@gmail.com

## Abstract

Although malaria is endemic in coastal Côte d'Ivoire, updated data on the resistance profile of the main vector, *Anopheles gambiae sensu lato* (s.l.), are still lacking, thus compromising decision-making for an effective vector control intervention. This study investigated the complex members and the insecticide resistance in the *Anopheles gambiae* s.l. populations in coastal Côte d'Ivoire. Between 2018 and 2020, cross sectional survey bioassays were conducted on female *An. gambiae* s.l. mosquitoes in three coastal health districts (Aboisso, Jacqueville and San Pedro) of Côte d'Ivoire. Pyrethroids deltamethrin, permethrin and alphacypermethrin (1X, 5X and 10X), clothianidin and synergist piperonyl butoxide (PBO) combined with pyrethroid 1X were tested using WHO tube bioassays. Chlorfenapyr was evaluated using CDC bottle bioassays. *An. gambiae* complex members and *kdr* 995F, *kdr* 995S and *Ace-1* 280S mutations were identified using polymerase chain reaction (PCR) technique. Overall, *An. gambiae* s.l. populations were primarily composed of *Anopheles coluzzii* (88.24%, n = 312), followed by *Anopheles gambiae sensu stricto* (7.56%) and hybrids (4.17%). These populations displayed strong resistance to pyrethroids at standard diagnostic doses, with mortality remaining below 98% even at 10X doses, except for alphacypermethrin in Aboisso. Pre-exposure to PBO significantly increased mortality but did not induce susceptibility, except for alphacypermethrin in Jacqueville. Clothianidin induced full susceptibility in Jacqueville and San Pedro, while chlorfenapyr induced susceptibility in Aboisso at 100 µg ai/bottle and all three districts at 200 µg ai/bottle. *kdr* 995F mutation dominated, with frequencies varying from 71.2% to 79.3%. *kdr* 995S had low, rates with frequencies ranging from 2.3% to 5.7%. *Ace-1* 280S prevalence varied between 4.2% and 42.9%. Coastal Côte d'Ivoire's *An. gambiae* s.l. populations were mainly composed of *An. coluzzii* and showed high resistance to pyrethroids. Clothianidin, chlorfenapyr, and PBO with

**Data Availability Statement:** All relevant data are within the manuscript.

**Funding:** The author(s) received no specific funding for this work.

**Competing interests:** The authors of this article formally declare the complete absence of any conflict of interest among them. Their commitment to integrity and transparency ensures an impartial contribution to the presented research. We declare that there is no conflict of interest regarding the publication of this paper. We certify that we have no financial or personal relationships with individuals or organizations that could inappropriately influence our work, and there is no professional or other personal interest of any nature or kind in any product, service, and/or company that could be construed as influencing the content of this paper.

pyrethroids increased mortality, indicating their potential use as an alternative for malaria vector control.

## Introduction

The World Health Organization (WHO) has recently reported that there were 249 million cases and 608,000 deaths from malaria in 2022 [1]. Malaria is most prevalent in sub-Saharan Africa, where the majority of cases and deaths occur. Within sub-Saharan Africa, countries such as Nigeria, the Democratic Republic of the Congo, and Mozambique have the highest malaria burdens. Malaria is one of the deadliest public health diseases among children less than 5 years old, particularly in sub-Saharan Africa. The WHO African region accounted for 95% of global malaria cases and 96% of deaths [2]. More than US$ 3.5 billion have been invested in 2021 for malaria, with a third of this investment (around US$ 1.1 billion) spent by the governments of malaria-endemic countries [2]. The WHO recommends malaria prevention strategies, including effective vector control tools that have major impacts in reducing the global burden of this disease. The WHO Global Technical Strategy (GTS) aims to reduce malaria incidence and mortality by at least 75% by 2025 and 90% by 2030, which seems to be challenging [2]. In 2021, 242 million Artemisinin-based Combination Therapies (ACTs) were distributed to the public health sector by National Malaria Control Programs (NMCPs) and about 590 million Insecticide Treated Nets (ITNs) were delivered to communities between2019 and2021 [2]. To achieve these goals, GTS has called for the development of new vector control tools that must incorporate new insecticide molecules (e.g., clothianidin and chlorfenapyr), synergists (e.g., pyperonyl butoxide: PBO) and/or insecticide mixtures containing at least two active ingredients with different modes of action to mitigate malaria vector resistance to insecticides [2–4]. Côte d'Ivoire's population lives in high malaria risk areas [5]. In 2021, malaria in the country was estimated at 7,443,146 cases with 14,906 deaths [2]. The main objectives of the National Control Malaria Program (NMCP) are to reduce malaria morbidity by 40% and the malaria mortality in high-burden by 33% by 2026 compared to the 2015 baseline [6]. The vector control strategy implemented by the NMCP is mainly based on the mass deployment of long-lasting insecticidal nets (LLINs) every three years and recently by implementing indoor residual sprays (IRS) [7–9] in one district. In Côte d'Ivoire, the National Malaria Strategic Plan 2016–2020 has prioritized indoor residual spraying (IRS) as an additional vector control method to reduce malaria morbidity and mortality [10]. The transmission of malaria in Côte d'Ivoire is via *An. gambiae* s.l., *An. funestus* s.l. and *An. nili* [7, 11]. *An. gambiae* s.l., or the *An. gambiae* complex, comprises nine species, including *An. gambiae sensu stricto* (s.s.), *An. coluzzii*, *An. arabiensis*, *An. melas*, *An. merus*, *An. bwambae*, *An. quadriannulatus*, *An. amharicus*, and *An. fontenillei*, which cannot be morphologically distinguished [12]. The species *An. gambiae* s.s., *An. coluzzii*, and the hybrids resulting from their interbreeding, all part of the *An. gambiae* complex, have been identified throughout the country [7, 13–16]. However, Ivorian *An. gambiae* s.l., populations exhibit strong resistance to many traditional classes of insecticides (i.e., DDT, pyrethroid, carbamate, and organophosphate) [7, 14–18]. Local *An. gambiae* s.l. resistance to insecticides is a severe challenge to the NMCP efforts because all LLINs distributed in Côte d'Ivoire before 2021 were treated with pyrethroid only [7, 10]. Therefore, it is urgent to develop effective alternative strategies for the sustainable control of insecticide-resistant malaria vectors. Since 2018, the National Malaria Strategic Plan (NMSP) supported by the U.S. President's Malaria Initiative (PMI) project has contributed to

the generation of insecticide resistance data to conduct a stratification of vector control interventions across the country. From 2018 to 2022, the NMCP and PMI deployed IRS in the health districts of Nassian and Sakassou using clothianidin-based insecticides that resulted in effective malaria vector control [6, 7]. Insecticide sensitivity tests have also been conducted so that these data can facilitate the development of mosquito nets incorporating pyrethroids and other active ingredients (synergists, pyrroles, etc.), as well as combinations of active ingredients, for the upcoming LLIN distribution campaign, based on entomological stratification data [6, 7]. Recently, pyrethroids in combination with either an insect growth regulator, a pyrrole, or a synergist that inhibits the primary metabolic mechanism of pyrethroid resistance within mosquitoes were used in LLINs [3, 4, 7]. In the current study, insecticide resistance status and complex members of *An. gambiae* s.l. were assessed in three coastal health districts of Côte d'Ivoire, namely Aboisso, Jacqueville and San Pedro. Although a low entomological inoculation rate (10.9 ib/p/y in Bardo) is recorded in the coastal zone of Côte d'Ivoire [11], this area has reported a high incidence of malaria. The health district of Jacqueville records one of the highest malaria incidences, three hundred sixty-seven per thousand (367‰), while a lower incidence is reported in the health district of San Pedro (127.1‰) [19, 20]. In Côte d'Ivoire, the coastal areas are characterized by a massive concentration of people [21]. These areas are the heart of the economic development of the country and have two main seaports, one in Abidjan and one in San-Pedro. This coastal region is home to the largest number of agro-industrial units, traditional agriculture and smallholdings [11, 15, 22]. The industrial and traditional crops (e.g., cocoa, rubber, oil palm, coffee, rice) [11, 13, 15, 16, 22] are intensively treated with pesticides and insecticides to protect crops [15]. The local *Anopheles* malaria vectors are constantly under strong pressure from insecticides and chemicals, which can lead to the development of their resistance and increased their survival. Thus there is an urgent need is to develop alternative insecticides and evaluate their effectiveness against malaria vectors in resistant areas, mainly on the coastline. The current study is part of a program designed to explore the landscape of malaria *Anopheles* vectors in coastal areas to inform disease control programs in Côte d'Ivoire.

## Methodology

### Study sites

The study was conducted in three health districts, Aboisso (latitude 5˚ 28' 04" N and longitude 3˚ 12' 25" W), Jacqueville (latitude 5˚ 12' 02" N and longitude 4˚ 24' 44" W) and San Pedro (latitude 4˚ 44' 5" N and longitude 6˚ 38' 10" W), located in the southern littoral area of Côte d'Ivoire (Fig 1). The local climate is characterized by two major seasons: the long rainy season from April to July, with the peak rainfall in June, and the long dry season from October to March. The annual rainfall average is 1848 mm with an annual temperature average of 27˚C. In the three health districts, economic activities are dominated by agriculture with large agricultural areas for food crops and cash crops (rubber, oil palm, pineapple, cassava, yam, banana) [21, 23, 24]. Farmers use several types of pesticides (herbicides, insecticides, and fungicides) to protect crops and increase production [15, 22, 25].

### Larval sampling and rearing to adults

Surveys for potential mosquito larvae habitats were conducted in different locations such as water puddles, vegetable cultivation sites, rice fields, and other potential larval habitats within our study sites. *Anopheles* mosquito larvae and pupae were collected using the dipping method [26] and combined. All collection site within each health district were combined. The larvae and pupae were collected from larval habitats between January 2018 and December 2020.

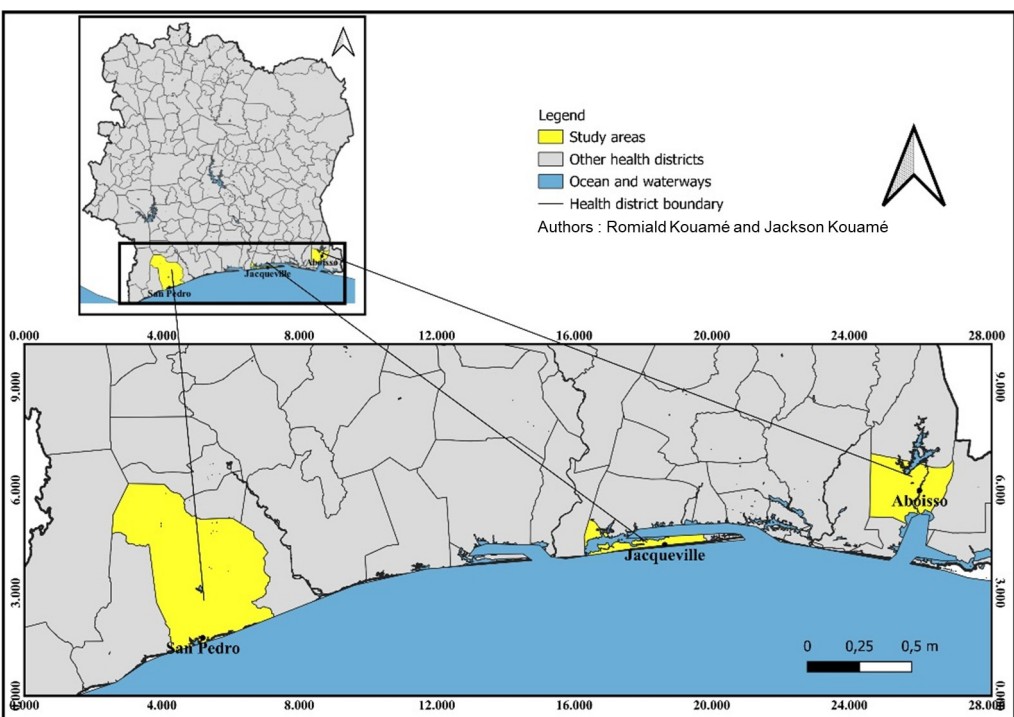

**Fig 1. Map of Côte d'Ivoire showing the location of the littoral health districts.** The map was generated using QGIS software version 3.32.3-Lima (https://www.qgis.org/), with the basemap shapefile sourced from the Database of Global Administrative Areas (GADM, https://gadm.org/; license: https://gadm.org/license.html).

Larvae were transferred to distilled water and reared into adults using the mosquito mass-rearing method [27] under standard laboratory conditions (27 ± 2∘C temperature; 70 ± 10% relative humidity; 12:12 hour light: dark photoperiod). Emerged adults were provided with a cotton wool pad soaked in a 10% sugar solution. Species (*An. gambiae* s.l.) were morphologically identified using identification keys [28] before being utilized for various sensitivity tests and moleculars analyses.

## WHO tube bioassay

The diagnostic dose tests of pyrethroids (alphacypermethrin, deltamethrin, and permethrin) were conducted in 2018, 2019, and 2020. The following tests, including intensity tests (1X, 5X, and 10X), synergist tests (PBO), and clothianidin tests, were exclusively conducted in 2020. Emerged adult females of *An. gambiae* s.l. were tested for insecticide susceptibility according to the WHO standard procedures [29]. Females of $F_0$ generations aged 2–5 days were used in all tests. Four batches of 20–25 non-blood-fed females were introduced each in four tube tests garnished with insecticide-treated filter papers, while two batches were exposed to control tubes with untreated filter papers for one hour. Mosquitoes were exposed to the WHO discriminating dosages of deltamethrin (0.05%), permethrin (0.75%) and alpha-cypermethrin (0.05%) to determine their resistance status (WHO, 2018). Moreover, synergist effects were assessed by pre-exposing mosquitoes to 4% PBO for one hour before being exposed to deltamethrin 0.05%, permethrin 0.75% and alphacypermethrin 0.05% for an additional hour. To determine the pyrethroid-resistance intensity, mosquitoes were exposed to 1X, 5X and 10X diagnostic concentrations of alpha-cypermethrin (0.05%, 0.25% and 0.5%), permethrin

(0.75%, 3.75% and 7.5%) and deltamethrin (0.05%, 0.25% and 0.5%). Mosquito mortality was recorded at 24 hours after exposure.

For clothianidin, impregnated papers were prepared in *situ* following the standard operating procedure (SOP) developed by the Vector Link entomology team. The diagnostic dose was set at 2% clothianidin. A solution of clothianidin was prepared by diluting 264 mg of the formulated product (Sumishield WG50) in 20 ml of distilled water. Four Whatman No.1 filter papers of 12 × 15 cm each were impregnated using a pipette to dispense 2 ml of solution on each filter paper, resulting in a concentration of 13.2 mg/ai clothianidin per paper. The untreated filter papers were treated using 2 ml of a solution of distilled water. Clothianidin was tested using WHO susceptibility tests, with the standard guidelines [30]. Mosquitoes were tested against clothianidin-treated papers as described above [31]. The mortality was recorded daily for up to seven days in order to capture any delayed mortality effects.

## CDC bottle test

The Centers of Disease Control and Prevention (CDC) bottle tests were exclusively conducted in 2020. Chlorfenapyr susceptibility was determined using CDC bottle tests. The CDC bottle tests utilized 250 ml glass bottles coated with 100 μg ai/bottle and 200 μg ai/bottle. A set of 15 to 20 non-blood-fed female *An. gambiae* s.l. aged 2–5 days were exposed to two discriminating concentrations of chlorfenapyr following the CDC bottle test protocol [32]. Mortality was recorded daily for up to three days.

## Molecular analyses of *An. gambiae* s.l.complex members and resistance genes

The *An. gambiae* complex members and *kdr* and *Ace-1* mutations are stored individually in Eppendorf tubes with silica gel, and maintained at −20˚C for subsequent identification.

## DNA extraction

The genomic DNA of individual mosquito was extracted according to the method described in Collins *et al.* [33]. Each mosquito was individually processed in a 1.5 ml Eppendorf tube, where it was soaked and crushed in 200 μl of 2% Cetyl Trimethyl Ammonium Bromide (CTAB), followed by incubation at 65˚C for 5 minutes. Afterwards, 200 μl of chloroform was added, and the resultant mixture was centrifuged at 12,000 rounds per minute (rpm) for 5 minutes. The supernatant was carefully transferred to a new 1.5 ml tube. Subsequently, 200 μl of isopropanol was added to the supernatant, thoroughly mixed by pipetting, and then centrifuged at 12,000 rpm for 15 minutes to precipitate the DNA. The DNA pellet formed at the bottom of the tubes and the supernatant was discarded. The DNA of each sample was purified with 200 μl of 70% Ethanol and centrifuged at 12,000 rpm for 5 minutes. After removing the ethanol, the pellet was air-dried overnight on the bench. Finally, the extracted DNA was reconstituted in 30 μl of DNase-free water (Sigma-Aldrich, United Kingdom), incubated at 65˚C for 15 minutes, and stored in the fridge at -20˚C.

## Complex member identification

The molecular identification of *An. gambiae* complex members was performed according to the SINE-PCR method [34], with minor modifications of reaction mixture. The PCR assay was performed using two primers, 6.1a (5'-TCGCCTTAGACCTTGCGTTA-3') and 6.1b (5'-CGCTTCAAGAATTCGAGATAC-3'). Each reaction mixture was done in a final volume of 25 μl containing 12.5 μl One taq Quick-load 2X Master Mix, 5.5 μl free water, 0.5 μl Dimethyl

sulphoxide (DMSO), 2 µl Bovine Serum Albumin (BSA), 1 µl of each primer, 0.5 µl Magnesium chloride (MgCl2) and 2 µl DNA template. The incubation took place in a thermocycler of LongGene® type (A200 Gradient Thermal cycler; LongGene Scientific Instruments Co., Ltd Hangzhou, P.R. China) according to the following program: 94 ∘C for 5 min, 94 ∘C for 25 sec, and 54 ∘C for 30 sec; 72 ∘C for 1 min repeated 35 times; and a final step at 72 ∘C for 10 min to terminate the reaction. After amplification, PCR products were run on either a 1.5% agarose gel in Tris/borate/EDTA (TBE) and stained with ethidium bromide solution for UV visualization. PCR products were loaded on the gel and allowed to migrate under a voltage of 140 V for an hour. The result was visualized with a UV illuminator (BioDoc- It Imaging System; Upland, CA, USA).

## Identification of target site mutation

The presence of insecticide resistance genes including *kdr*-L995F (previously known as *kdr*-L1014F (*kdr-West*)), *kdr*-L995S (previously known as *kdr*-L1014S (*kdr-East*)), and *Ace-1*-G280S (previously known as *Ace-1*-G119S) [35] was investigated using real-time PCR. The primers *Kdr*-Forward (5'-CATTTTTCTTGGCCACTGTAGTGAT-3') and *Kdr*-Reverse (5'-CGATCTTGGTCCATGTTAATTTGCA-3') were standard oligonucleotides without any modifications. The probe WT (5'-CTTACGACTAAATTTC-3'), labeled with HEX at the 5' end, was used for detecting the wildtype allele. The probes *kdr* W (5'-ACGACAAAATTTC-3') and *kdr* E (5'-ACGACTGAATTTC-3') were labeled with FAM for detecting the *kdr*-W and *kdr*-E alleles, respectively. Similarly, the primers *Ace-1*-Forward (5'-GGCCGTCATGCTGTGGAT-3') and *Ace-1*-Reverse (5'-GCGGTGCCGGAGTAGA-3') were also standard oligonucleotides. The probe WT (5'-TTCGGCGGCGGCT-3') was labeled with HEX, while the probe (5'-TTCGGCGGCAGCT-3') was labeled with FAM. The TaqMan assays as detailed by Bass [36] were used to screen for the 995F, 995S *kdr* and 280S mutations. The reaction was carried out in an Agilent Stratagene MX3005 qPCR thermocycler (Agilent Technologies, Santa Clara, CA, USA). Each 1 µl of gene DNA was combined with a total volume of 9 µl of master mix, comprising 3.875 µl of DNase-free water, 5 µl of SensiMix, and 0.125 µl of specific primer/probe for either *kdr* 995F, 995S, or *Ace-1*-280S. *KdrAce-1* The specific probe contains FAM and HEX fluorochromes. FAM was used to detect the mutant allele, while HEX detected the wild-type susceptible allele. PCR conditions used were 10 min at 95 ∘C (1 cycle), followed by 40 cycles of 10 sec at 95 ∘C, and 45 sec at 60 ∘C.

## Data analysis

All data were entered in Excel. R software version 4.2.1 (2022-06-23 ucrt) was used for the various statistical analyses. The resistance status of each mosquito population was determined according to the WHO criteria with mortality after 24-hour, 72-hour and 7-day post-exposure for pyrethroids, chlorfenapyr and clothianidin, respectively [37]. Mortality was corrected using Abbott's formula (Corrected mortality $= \frac{\text{(Mortality treated−mortality control)}}{\text{(100−mortality of untreated control)}}$ X100), when the mortality of the control tubes was above 5% and less than 20% [38]. There was a confirmed resistance if the mortality percentage was <90%, possible resistance if the mortality rate was between 90 and 98%, and susceptibility if the mortality rate was ≥98%. The package 'lsmeans' version 2.30–0 was used for various analyses. The mortality recorded each year was compared between each insecticide (pyrethroid diagnostic concentration) using the pairwise tests of the generalized linear model (GLM). For the resistance intensity, if corrected mortality was 98–100% at 5X the diagnostic dose indicated low resistance intensity. However, if mortality was less than 98% at 5X diagnostic dose implied testing the 10X diagnostic dose.At 10 times the diagnostic dose, a corrected mortality rate of 98–100% confirms a moderate resistance

intensity.; A corrected mortality rate of less than 98% at 10 X the diagnostic dose indicates a high level of resistance. For the synergist assays, the mortality of mosquitoes exposed to PBO with pyrethroid was compared with that of the insecticides without PBO pre-exposure. Comparison was made between mortality rates with and without PBO pre-exposure using the *prop. test* with software R version 4.2.1. The frequency of resistance mutations (*kdr*-995F, *kdr*-995S and *Ace-1* 280S) was determined using the formula: F = [(2AA+ Aa)] / [2(AA+Aa+aa)] [39] with aa, homozygous susceptible genotype; Aa, heterozygous genotype; AA, homozygous resistant genotype.

## Results

### Species composition of the *Anopheles gambiae* complex

A total of 672 *An. gambiae* s.l. were selected from the three sites for species identification. Overall, *An. gambiae* complex was mainly composed of *An. coluzzii* (88.2%, n = 593), followed by *An. gambiae* s.s. (7.6%, n = 51) and hybrids (*An. coluzzii/An. gambiae* s.s.) (4.2%, n = 27). In Jacqueville and San Pedro, all mosquitoes were identified as *An. coluzzii*. The two species were found in sympatry in Aboisso. However, *An. coluzzii* dominated in Aboisso (66.7%, n = 158), followed by *An. gambiae* s.s. (21.5%, n = 51) and hybrids (11.8%, n = 28) (Table 1).

### Insecticide susceptibility in *An. gambiae* s.l.

Table 2 presents the mortality rates of *An. gambiae* s.l. populations from the surveyed sites (Aboisso, Jacqueville, and San Pedro) when exposed to pyrethroids (alphacypermethrin 0.05%, permethrin 0.75%, and deltamethrin 0.05%) over the years 2018, 2019, and 2020. Across all sites, the mortality rates for all three pyrethroids remained below 30%, confirming significant resistance of *An. gambiae* s.l. to these insecticides.

Table 3 summarizes the comparison of insecticide-induced mortality rates by site and year. In 2020, a statistically significant difference was observed in Aboisso between mortality rates due to deltamethrin and alpha-cypermethrin (estimate: 2.45; Z-ratio: 3.25; p-value = 0.0033). Similarly, in 2020, permethrin also showed a statistically significant difference compared to alpha-cypermethrin in Aboisso (estimate: 3.42; Z-ratio: 3.32; p-value = 0.0026). In Jacqueville, the 2020 comparison between permethrin and alpha-cypermethrin revealed a statistically significant difference (estimate: 2.08; Z ratio: 2.73; p-value = 0.0174).

### Intensity of resistance

The results of the pyrethroid intensity test are summarized in Fig 2. These results confirmed the strong resistance of *An. gambiae* s.l. against alpha-cypermethrin, permethrin and deltamethrin. In the resistance pyrethroid intensity tests, conducted on samples from Aboisso, Jacqueville, and San Pedro, mortalities remained below the 98% threshold when exposed to 1X and

**Table 1. Distribution of *Anopheles gambiae* s.l. in the three studied site.**

| Population | Species | Detected (N tested) | Percentage (%) |
|---|---|---|---|
| Aboisso | *An. coluzzii* | 158(237) | 66,7 |
| Aboisso | Hybrid (*An. coluzzii/An. gambiae* s.s.) | 28(237) | 11,8 |
| Aboisso | *An. gambiae* s.s. | 51(237) | 21,5 |
| Jacqueville | *An. coluzzii* | 221(221) | 100 |
| San Pedro | *An. coluzzii* | 214(214) | 100 |

N: Number of mosquitoes; %: Percentage

**Table 2. Mortality observed after 24 h with pyrethroid insecticides in *Anopheles gambiae* s.l. from 2018 to 2020.**

| Populations | years | Insecticide | N tested (Dead) | Mortality percentage (95% CI) |
|---|---|---|---|---|
| **Aboisso** | 2018 | deltamethrin | 190(10) | 5,26 (2.70–9.74) |
| | | alphacypermethrin | 93(0) | 0(0–4,94) |
| | | permethrin | 101(06) | 5,94(2.43–12.99) |
| | 2019 | deltamethrin | 84(5) | 5,95(2.21–13.96) |
| | | alphacypermethrin | 81(0) | 0(0–4.34) |
| | | permethrin | 84(0) | 0(0–5.64) |
| | 2020 | deltamethrin | 79(2) | 2.53(0.44–9.69) |
| | | alphacypermethrin | 82(24) | 29.27(20–40.50) |
| | | permethrin | 105(1) | 0.95 (0.05–5.95) |
| **Jacqueville** | 2018 | deltamethrin | 93(5) | 5,37(1.20–12,67) |
| | | alphacypermethrin | 90(0) | 0(0–5,10) |
| | | permethrin | 94(2) | 2,12(0.37–8,21) |
| | 2020 | deltamethrin | 90(10) | 11,11(5.75–19,92) |
| | | alphacypermethrin | 89(18) | 20,22(12.74–33.34) |
| | | permethrin | 79(2) | 2.53(0.44–9.69) |
| **San Pedro** | 2018 | deltamethrin | 100(4) | 4(1.29–10.51) |
| | | alphacypermethrin | 99(0) | 0(0–4.65) |
| | | permethrin | 98(2) | 2.04(0.35–7.89) |
| | 2019 | deltamethrin | 195(7) | 3.59(1.58–7.55) |
| | | alphacypermethrin | 94(0) | 0(0–4.89) |
| | | permethrin | 94(4) | 4,25(1.37–11.15) |
| | 2020 | deltamethrin | 94(0) | 0(0–4.89) |
| | | alphacypermethrin | 101(0) | 0(0–4.57) |
| | | permethrin | 93(1) | 1,08(0.06–6.69) |

5X concentrations of all insecticides(Fig 2). However, at a concentration of 10X, a high intensity of resistance was still observed across all three mosquito populations for the insecticides, except for a case of moderate intensity resistance recorded with alphacypermethrin 0.5% in the Aboisso populations, resulting in 100% mortality.

## Effect of piperonyl butoxide (PBO)

Table 4 shows the mortality rates in the Aboisso, Jacqueville and San Pedro *An. gambiae* s.l. populations exposed to alpha-cypermethrin, permethrin and deltamethrin without and with pre-exposure to PBO. The results showed that synergist effect of PBO was strongest in all mosquito populations. Mortality increased significantly when mosquitoes were pre-exposed to PBO (df = 1, p <0.0001) (Table 1). Although a significant increase was observed in mortality after pre-exposure to PBO, susceptibility was not fully restored. The mortality rates after PBO pre-exposure were still under the 98% threshold in all three study sites, except for the Jacqueville populations in which alphacypermethrin mortality after pre-exposure to PBO was 98.5% (Table 4).

## Effect of clothianidin and chlorfenapyr

After seven days of observation, *An. gambiae* s.l. populations from Jacqueville and San Pedro were found to be completely susceptible to clothianidin (Fig 3). The San Pedro population achieved a 100% mortality rate by the sixth day (Fig 3). In Jacqueville, the sensitivity threshold was reached on the seventh day, with a mortality rate of 98.8%. However, in Aboisso,

**Table 3. Generalized linear model showing the difference in insecticide mortality among *Anopheles gambiae* s.l. populations along the coastline from 2018 to 2020.**

| Populations | Years | Comparison between insecticide | estimate | Z-ratio | p-value |
|---|---|---|---|---|---|
| **Aboisso** | 2018 | deltamethrin vs alphacypermethrin | -18.94 | -0.005 | 1 |
| | | alphacypermethrin vs permethrin | -19.07 | -0.005 | 1 |
| | | permethrin vs deltamethrin | -0.121 | -0.228 | 0,97 |
| | 2019 | deltamethrin vs alphacypermethrin | -20.94 | -0.002 | 1 |
| | | alphacypermethrin vs permethrin | 0.03 | 0 | 1 |
| | | permethrin vs deltamethrin | 20.97 | 0.002 | 1 |
| | 2020 | deltamethrin vs alphacypermethrin | 2.45 | 3.252 | 0.0033 |
| | | alphacypermethrin vs permethrin | 3.42 | 3.322 | 0.0026 |
| | | permethrin vs deltamethrin | 0.98 | 0.792 | 0.7077 |
| **Jacqueville** | 2018 | deltamethrin vs alphacypermethrin | -19.93 | -0.003 | 1 |
| | | alphacypermethrin vs permethrin | -19.012 | -0.003 | 1 |
| | | permethrin vs deltamethrin | 0.92 | 1.091 | 0.5194 |
| | 2020 | deltamethrin vs alphacypermethrin | 0.59 | 1.420 | 0.3305 |
| | | alphacypermethrin vs permethrin | 2.08 | 2.730 | 0.0174 |
| | | permethrin vs deltamethrin | 1.47 | 1.873 | 0.1466 |
| **San Pedro** | 2018 | deltamethrin vs alphacypermethrin | -19.73 | -0.003 | 1 |
| | | alphacypermethrin vs permethrin | -19.05 | -0.003 | 1 |
| | | permethrin vs deltamethrin | 0.67 | 0.767 | 0.7234 |
| | 2019 | deltamethrin vs alphacypermethrin | -18.58 | -0.005 | 1 |
| | | alphacypermethrin vs permethrin | -18.75 | -0.005 | 1 |
| | | permethrin vs deltamethrin | -0.17 | -0.266 | 0.9617 |
| | 2020 | deltamethrin vs alphacypermethrin | -0.06 | 0.000 | 1 |
| | | alphacypermethrin vs permethrin | -20.4 | -0.001 | 1 |
| | | permethrin vs deltamethrin | -20.36 | -0.001 | 1 |

susceptibility was not fully restored after seven days of post-exposure observation, with the mortality rate only reaching 55.7%.

For chlorfenapyr at 100 µg/bottle, the Aboisso population was fully susceptible, with a 72-hour mortality of 100%. However, chlorfenapyr did not fully induce susceptibility in Jacqueville and San Pedro populations with 100 µg/bottle. With 100 µg/bottle, the 72-hour mortality was 88.7% for Jacqueville and 97.1% for San Pedro populations (Fig 4). With 200 µg/ bottle, the mortality rate was 100% for the San Pedro population after 24-hours post-exposure. Except for the Jacqueville population where 100% mortality was recorded after 48-hours post-exposure (Fig 4). The population of Aboisso was not tested at 200 µg/bottle, because susceptibility was achieved at 100 µg/bottle.

## Resistance mutation

Table 5 shows the *kdr* (L995F and L995S) mutation allelic frequencies in the population from Aboisso, Jacqueville and San Pedro (Table 2). The *kdr* L995F mutation allelic frequencies was from 79.3% for *An. coluzzii*, 76.0% for hybrids, and 72.1% for *An. gambiae* s.s. in the Aboisso population. In the population from Jacqueville and San Pedro, the *kdr* L995F (*kdr west*) frequencies were respectively 38.1% and 71.2%. For the *kdr* L995S (*kdr East*), the mutation allelic frequencies were low. The mutation allelic frequencies were 2.3% for *An. gambiae* s.s. in Aboisso population and 5.7% in San Pedro population. For the population of Aboisso, and San Pedro mutation was not detected. The mutation allelic frequencies of G280S were summarized in Table 5. The mutation allelic frequencies of G280S were 27.4% for *An. coluzzii*, and 42.9%

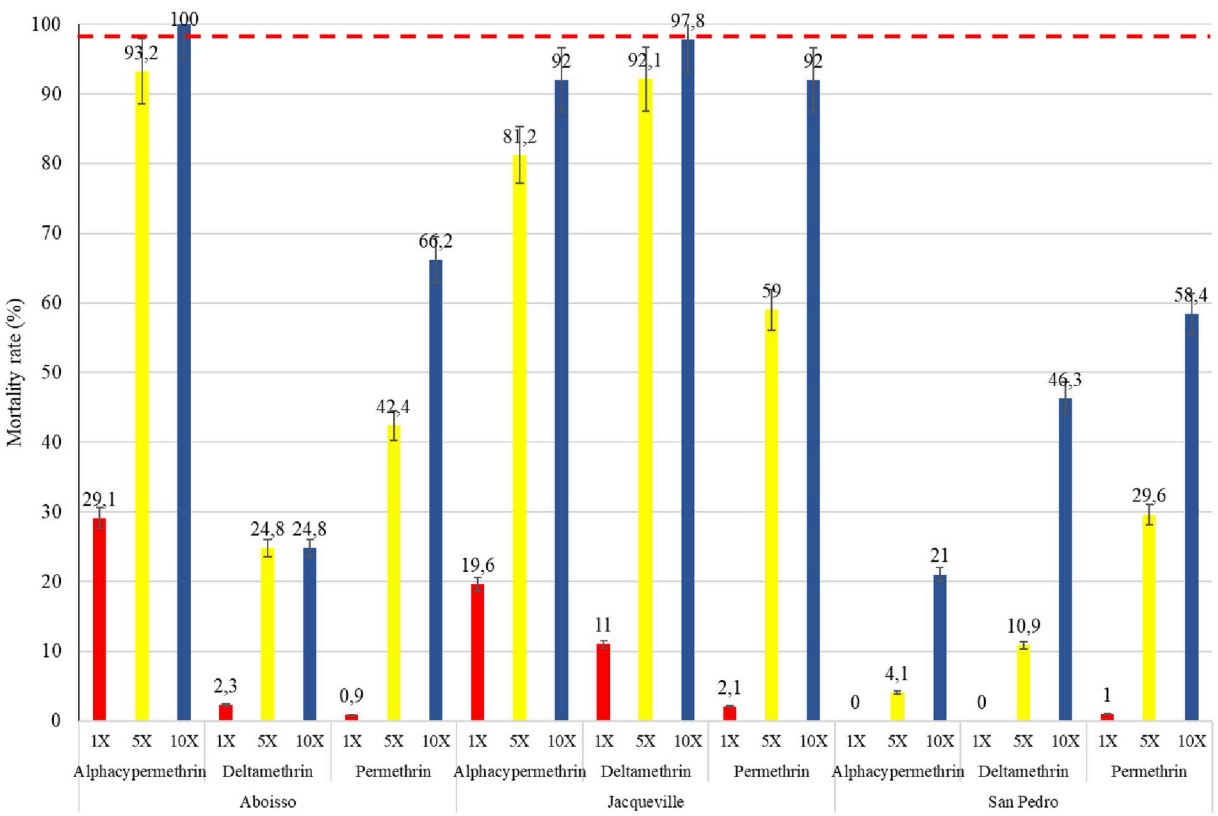

**Fig 2. Mortality rates of *Anopheles gambiae* s.l. population exposed to pyrethroid intensity tests from the three study populations.** The red dashed line represents the susceptibility threshold.

for *An. gambiae* s.s. in the population of Aboisso. The allelic frequencies were 31.1% and 4.2% respectively for the population of Jacqueville and San Pedro.

## Discussion

This study investigated the *An. gambiae* complex members resistance levels and mechanisms to pyrethroids (i.e., alpha-cypermethrin permethrin and deltamethrin) with and without PBO, clothianidin and chlorfenapyr in three coastal health districts, Aboisso, Jacqueville and San Pedro in southern Côte d'Ivoire. To our knowledge, this is one of the rare studies investigating both, the resistance status and complex members of malaria *Anopheles* mosquitoes in Costal Côte d'Ivoire. The current study showed that *An. gambiae* s.l. was composed of two sibling species, namely *An. coluzzii*, *An. gambiae* s.s, and the hybrids. Among the two sibling species *An. coluzzii* was the predominant (88.24%) species. The samples from Jacqueville and San Pedro were composed only of *An. coluzzii*. Both sibling species were represented in the Aboisso *An. gambiae* s.l. population. Among the sibling species, *An. coluzzii* (66.95%) was most abundant, followed by *An. gambiae* s.s. and hybrids. Moreover, *An. gambiae* s.l. showed high phenotypic and genotypic resistance to pyrethroids, with increased mortality while pre-exposed to PBO. Overall, *An. gambiae* s.l. susceptibility was fully reestablished by clothianidin and chlorfenapyr throughout, except for the Aboisso population with clothianidin. *Kdr* 995F allele frequency was very high (0.72–0.80%) in all *An. gambiae* s.l. populations. *kdr* 995S was very low (0–0.06%) in all *An. gambiae* s.l. populations. For *Ace-1* 280S, the high frequency (0.57%) was recorded in San Pedro and the low (0) from the Aboisso hybrid population.

**Table 4. Mortality of An. gambiae s.l. to pyrethroids without and with piperonyl butoxide of the study population.**

| Populations | Insecticide | N tested (Dead) | Mortality percentage (95% CI) | p-value |
|---|---|---|---|---|
| **Aboisso** | Alphacypermethrin | 82 (24) | 29.1 (19.73–40.35) | <0.0001 |
| | PBO+Alphacypermethrin | 76 (72) | 94.7 (87.07–98.55) | |
| | Deltamethrin | 79 (2) | 2.3 (0.31–8.85) | <0.0001 |
| | PBO + Deltamethrin | 84 (42) | 50.0 (38.88–61.11) | |
| | Permethrin | 105 (1) | 0.9 (0.02–5.19) | <0.0001 |
| | PBO + Permethrin | 86 (28) | 32.5 (22.84–43.52) | |
| **Jacqueville** | Alphacypermethrin | 89 (18) | 19.6 (12.45–30.07) | <0.0001 |
| | PBO+Alphacypermethrin | 82 (74) | 90.1 (81.68–95.69) | |
| | Deltamethrin | 90 (10) | 0 (5.46–19.48) | <0.0001 |
| | PBO + Deltamethrin | 71 (70) | 98.5 (92.40–99.96) | |
| | Permethrin | 79 (2) | 2.1 (0.31–8.85) | <0.0001 |
| | PBO + Permethrin | 68 (22) | 31.7 (21.51–44.79) | |
| **San Pedro** | Alphacypermethrin | 101 (0) | 0 (0–3.59) | <0.0001 |
| | PBO+Alphacypermethrin | 106 (19) | 17.9 (11.15–26.57) | |
| | Deltamethrin | 94 (0) | 0 (0–3.85) | <0.0001 |
| | PBO + Deltamethrin | 98 (28) | 28.3 (19.90–38.58) | |
| | Permethrin | 93 (1) | 1.1 (0.03–05.85) | <0.0001 |
| | PBO + Permethrin | 103 (11) | 10.6 (5.45–18.31) | |

N = represents the total number of mosquitos tested, CI = represents the confidence intervals, and p-value were calculated using the prop.test.

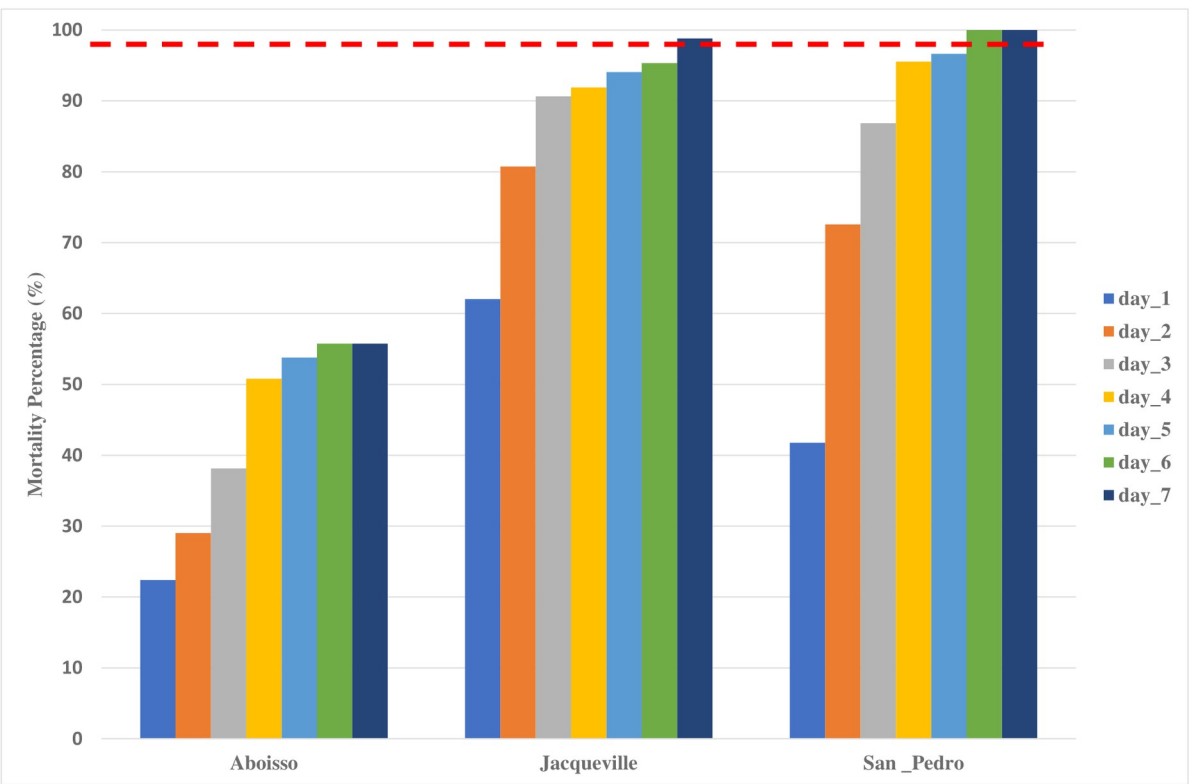

**Fig 3. Mortality of *Anopheles gambiae* s.l. exposed to 2% clothianidin in 2020 from the three study populations.** The red dashed line represents the susceptibility treshold.

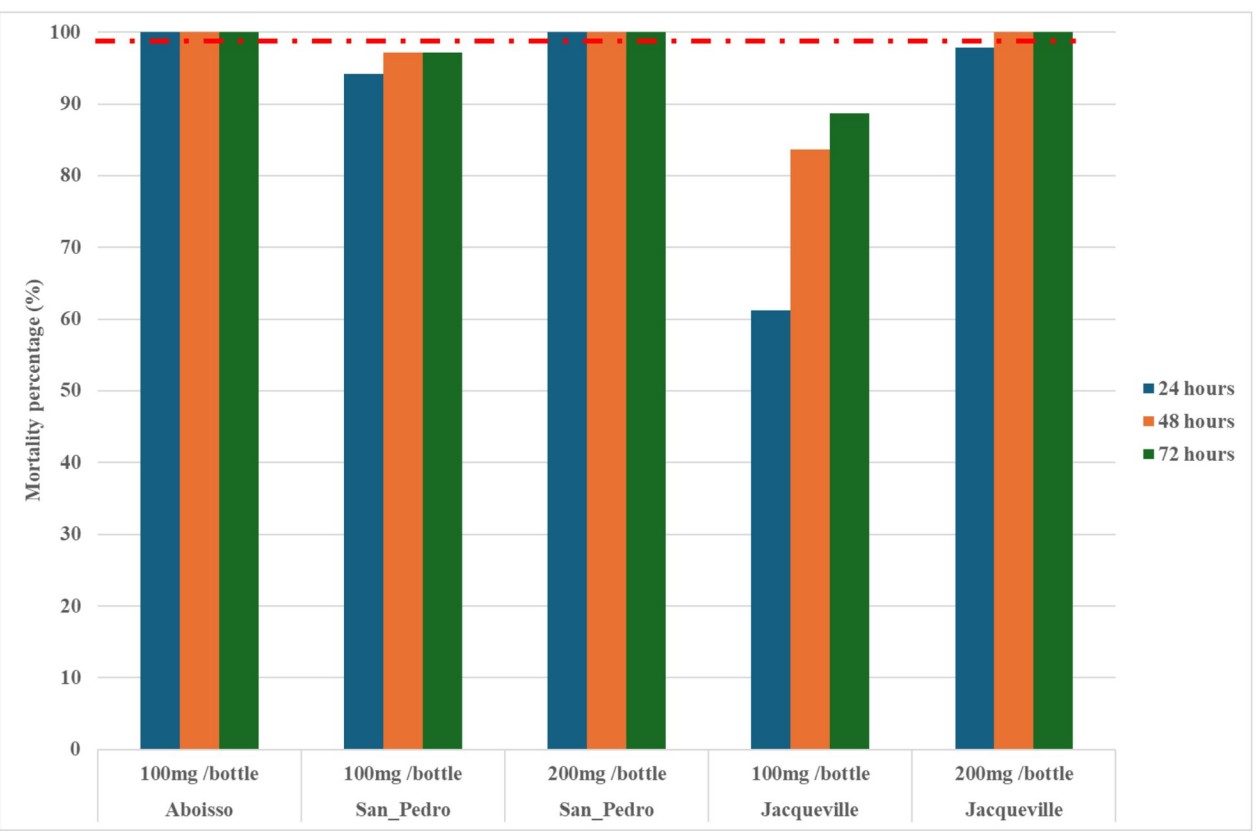

**Fig 4. Mortality of *Anopheles gambiae* s.l. exposed to chlorfenapyr in CDC bottle bioassays in 2020 from the three study populations.** The red dashed line represents the susceptibility treshold.

Species identification of *An. gambiae* s.l. showed that *An. coluzzii* constituted the major species among the *An. gambiae* complex. In San Pedro and Jacqueville, only the species *An. coluzzii* was identified. This selective distribution of vectors is of considerable importance for understanding malaria transmission in these regions. *Anopheles coluzzii's* preference for hosts and specific breeding habitats, especially its attraction to the littoral region, may account for its dominance in these zones [40]. Furthermore, the geographical distribution of various *An. gambiae* s.l. species may result from local ecological variations and adaptations to specific microclimates at each site. In Aboisso, the dominance of *An. coluzzii* over *An. gambiae* s.s. and

**Table 5. An. gambiae s.l. sibling species composition and kdr-995F, kdr-995S and Ace-1 280S frequency in three populations.**

| Populations | *An. gambiae* s.l. molecular form | kdr-West (L995F) | | | | | kdr-East (L995S) | | | | | Ace-1 (G280S) | | | | |
|---|---|---|---|---|---|---|---|---|---|---|---|---|---|---|---|---|
| | | Tested | FF | LF | LL | %Fr (F) | Tested | SS | LS | LL | %Fr (S) | Tested | SS | GS | GG | %Fr (S) |
| **Aboisso** | ***An. coluzzii*** | **127** | **83** | **26** | **12** | **79.3** | **127** | **0** | **0** | **127** | **0** | **31** | **5** | **7** | **19** | **27.4** |
| **Aboisso** | Hibryd | 25 | 16 | 6 | 3 | 76.0 | 25 | 0 | 0 | 25 | 0 | 2 | 0 | 0 | 2 | 0 |
| **Aboisso** | *An. gambiae* s.s. | 43 | 23 | 16 | 4 | 72.1 | 44 | 0 | 2 | 42 | 2.3 | 7 | 3 | 0 | 4 | 42.9 |
| **Jacqueville** | *An. coluzzii* | 145 | 86 | 45 | 14 | 38.1 | 140 | 1 | 14 | 125 | 5.7 | 82 | 4 | 43 | 35 | 31.1 |
| **San Pedro** | *An. coluzzii* | 99 | 57 | 27 | 15 | 71.2 | 100 | 0 | 0 | 100 | 0 | 108 | 0 | 9 | 99 | 4.2 |

LL, and GG homozygous susceptible genotype (aa); LF, LS, and GS heterozygous genotype (Aa); FF, and SS, homozygous resistant genotype (AA). The frequencies of resistance mutations were calculated using the formula: F = [(2AA+ Aa)] / [2(AA+Aa+aa)].

crossbreeds carries profound implications for malaria transmission and vector control [41]. *Anopheles coluzzii* prevalence underscores its role as the primary vector, given its affinity for human hosts and urban breeding sites, which enhances the efficiency of malaria transmission [42]. The lower prevalence of *An. gambiae* s.s. may be attributed to ecological factors or competition with *An. coluzzii* [13]. Hybrid mosquitoes, comprising 11.8% of the complex, introduce variables such as insecticide resistance and altered behavior [43, 44]. Customizing vector control strategies, including the use of insecticide-treated bed nets and indoor spraying, to the dominant species, particularly *An. coluzzii*, is crucial [45]. The dominance of *An. coluzzii* species among *An. gambiae* complex is commonly reported in southern Côte d'Ivoire [7, 11, 13, 15, 18, 22]. Human activities such as urban Agriculture (vegetable gardens, irrigated rice fields) could have contributed to the development of the breeding sites of *An. gambiae* s.l. [25, 46]. The presence of sibling species, namely *An. coluzzii*, *An. gambiae* s.s. and hybrids, which is consistent with previous studies in the same area [13, 18]. Along with the presence of both *An. coluzzii* and *An. gambiae* s.s., and hybridsin the same area found here may imply that the mating appears to be occurring between the two species.

In the bioassays conducted in this study with pyrethroid, the highest mortalities were obtained with deltamethrin. Indeed, deltamethrin and alphacypermethrin are type II pyrethroids contrary to permethrin which is type I. Type II pyrethroids are distinguished from type I pyrethroids by the presence of a cyano group at the carbon of the esterified alcohol [47]. The effect of the cyano group on insecticidal activity increases mortality [48]. In such settings, use of deltamethrin or alphacypermethrin would probably be a good choice to treat nets. Several studies have shown the performance of type II pyrethroids in comparison to Type I [49]. However, the mortality varied between both insecticides. Several reasons would explain deferred mortalities induced by the same insecticide in different years. Pyrethroid resistance was detected more than 30 years ago in malaria vectors (*An. gambiae* s.l.) in Côte d'Ivoire for the first time in vectors from the Bouaké population [50]. In this study, *An. gambiae* s.l. from Aboisso, Jacqueville and San Pedro showed very strong phenotypic resistance to pyrethroids resulting in a very low mortality at standard diagnostic doses. The result is consistent with the general trends of *An. gambiae* s.l. resistance reported by previous studies in Côte d'Ivoire [16, 18] and reinforces the view that Côte d'Ivoire is a hotspot of resistance in West Africa [14, 16, 18]. Phenotypic resistance to pyrethroids may indicate the presence of genetic resistance resulting from modifications to the target sites of the insecticides. This could be linked to the extensive use of the same insecticides in both agricultural and public health contexts. Additionally, other factors such as the inherent genetic variability of insect populations and agricultural practices may also contribute to the emergence and spread of resistance [22–24]. The coastal zones in southern Côte d'Ivoire are devoted to large agro-industrial units (cocoa, rubber and oil palm plantations [21, 23, 51]) which are treated with chemicals, including insecticides, to protect the crops [14, 33–35, 52]. Indeed, the massive and incorrect use of pesticides by untrained farmers can contribute to the development of multiple resistance mutations in *An. gambiae* s.l. against insecticides used in public health [53].

In this study, very high allelic frequencies of the L995F *kdr* gene (>38%) were found in *An. gambiae* s.l. from the coastal health districts of Aboisso, Jacqueville, and San Pedro. These elevated allele frequencies indicate the widespread presence of the L995F and *kdr-West* mutations within the local *An. gambiae* s.l. population, as observed in several regions of Côte d'Ivoires.l. [12, 14, 15]. The low allelic frequencies of the L995S *kdr* gene (<6%) were detected in local *An. gambiae* s.l. The 995S allele was only found in *An. gambiae* s.s. from Aboisso and *An. coluzzii* from Jacqueville. Nonetheless, the presence of this allele can deliver further evidence for a pan-African propagation of the *kdr* resistance. This spread of the *kdr* 995F and 995S gene in *An. gambiae* s.l. populations in littoral areas could compromise the effectiveness of the vector

control tools that are currently in use. The *kdr* genes are known to confer resistance to pyrethroid insecticides, which are commonly used in long-lasting insecticidal nets (LLINs) and indoor residual spraying (IRS). If the *kdr* 995F and 995S mutations spread, the effectiveness of these malaria vector control tools (LLINs and IRS) would be compromised [54]. The high frequencies of the G119S mutation in Aboisso and Jacqueville suggest its widespread presence in the coastal areas of Côte d'Ivoire. *Ace-1* G119S allele confers resistance to organophosphates and carbamates that are commonly used in Côte d'Ivoire for crop protection [14, 35].

The present study demonstrates that increased insecticide concentrations and PBO pre-exposure significantly heightened mortality in all three coastal populations of *An. gambiae* s.l. against all pyrethroids. However, this approach did not fully achieve susceptibility. Indeed, the current study showed that high intensity resistance is present in all three *An. gambiae* s.l. populations, with overall mortality being under 98% at 1X, 5X and 10X for all pyrethroid insecticides as reported by Kouassi [18] who carried out similar tests in Southern and central districts of Cote d'Ivoire. Reduced insecticidal activity can result in lower personal protection by allowing more insects to survive [55]. This increases the likelihood of bites and the spread of insect-borne diseases, thereby threatening the effectiveness of LLINs [32]. According to WHO, vector control operational failure is likely to happen if resistance is confirmed at the 5X and especially at the 10X concentrations [32]. The various anthropogenic or natural xenobiotics present in local mosquito breeding sites and the biotic interaction could lead to the development of very high levels of resistance observed in the study [56]. The high intensity resistance recorded using only pyrethroid could be explained by high *kdr* mutation frequency, mixed function oxidases (MFO) activities, and an increase in esterase activity. However, moderate resistance intensity was recorded with a 10X concentration of alphacypermethrin in the Aboisso populations, and additional investigations are required for better understanding. Although insecticide-induced mortality increased, full sensitivity with PBO was only observed in the Jacqueville strain with deltamethrin. However, full susceptibility was not recovered using PBO, except with the strain of Jacqueville with deltamethrin. Indeed, mortalities in PBO-pre-exposed mosquitoes tested against pyrethroids were still lower than the WHO susceptibility cut-off, as observed in other areas in Côte d'Ivoire [18, 57]. Significant increase in mortality after exposure to PBO may indicate the presence of metabolic resistance (MFO, P450s, and esterases) [57], and this can compromise the effectiveness of non-PBO LLINs (e.g., Panda.Net, MagNet, PermaNet 2.0). In addition, non-restoration of full susceptibility with PBO might reduce the efficacy of PBO-treated LLINs (e.g., PermaNet 3.0) [58].

In contrast to PBO, chlorfenapyr and clothianidin reestablished fully the susceptibility in all populations of *An. gambiae* s.l. from Aboisso, Jacqueville and San Pedro, except for clothianidin in Aboisso population in the current study. The chlorfenapyr (pyrrole) and clothianidin (neonicotinoid) have new modes of action and are good candidates for malaria vector control in areas comprised of mutation and/or metabolic resistance to pyrethroids [59]. Indeed, after exposure to chlorfenapyr, full susceptibility was recovered in one of three districts at 100μg ai/bottle. However, when the samples were tested with higher dosages (200μg ai/bottle), the mortality increased, and susceptibility was achieved in all districts. Chlorfenapyr requires activation, via cytochrome P450 monooxygenases [60]. Chlorfenapyr-treated LLINs (e.g., Interceptor G2 and PermaNet Dual) could be effective against *An. gambiae* s.l. in coastal Côte d'Ivoire [3, 4, 59]. Moreover, *An. gambiae* s.l. susceptibility was achieved with clothianidin (100% mortality) in Jacqueville and San Pedro. As the National Malaria Strategic Plan of the National Malaria Control Programme of Côte d'Ivoire prioritizes indoor residual spraying (IRS), using clothianidin-based IRS (e.g., Fludora® Fusion) could be effective against the coastal *An. gambiae* s.l. populations in coastal Côte d'Ivoire.

## Conclusion

The present study showed that *An. gambiae* s.l. was predominate by *An. coluzzii*, followed by *An. gambiae* s.s and hybrids in the three studied sites of costal Côte d'Ivoire. Local *An. gambiae* s.l. populations were highly resistant to pyrethroids and susceptible to clothianidin and chlorfenapyr. Pre-exposure to PBO increased the mortality, but susceptibility was not fully recovered. The coastal *An. gambiae* s.l. populations had high frequencies of target site mutation genes (*kdr*-West and *kdr*-East) and metabolic genes (*Ace-1*). *An. gambiae* s.l. proved fully susceptible to clothianidin and chlorfenapyr. This suggests that vector control tools, such as insecticide-treated nets and sprays, treated with these new insecticides could be highly effective in reducing malaria transmission in coastal Côte d'Ivoire. Their use could significantly improve malaria control and reduce the disease burden in the region.

## Supporting information

**S1 Table. Data base assessing species composition and insecticide resistance of *Anopheles gambiae* complex members in three coastal health districts of Côte d'Ivoire.**
(XLSX)

## Acknowledgments

We gratefully acknowledge the contribution of technicians Assamoi Jean Baptiste, Bamogo Kassoum and Dobri Didier of the Centre Suisse de Recherches Scientifiques en Côte d'Ivoire for help and assistance during laboratory analysis. We also thank the local health district staff and a special thanks to the Programme National de Lutte contre le Paludisme Côte d'Ivoire for facilitating and participating in field data collection.

## Author Contributions

**Conceptualization:** Jackson K. I. Kouamé, Constant V. A. Edi, Benjamin G. Koudou.

**Data curation:** Jackson K. I. Kouamé.

**Formal analysis:** Jackson K. I. Kouamé, Ruth M. A. Kouamé.

**Investigation:** Jackson K. I. Kouamé, Yves A. K. Kacou.

**Methodology:** Jackson K. I. Kouamé, Constant V. A. Edi, Julien B. Z. Zahouli, Benjamin G. Koudou.

**Software:** Jackson K. I. Kouamé.

**Supervision:** Constant V. A. Edi, Benjamin G. Koudou.

**Visualization:** Firmain N. Yokoly.

**Writing – original draft:** Jackson K. I. Kouamé.

**Writing – review & editing:** Jackson K. I. Kouamé, Julien B. Z. Zahouli, Ruth M. A. Kouamé, Firmain N. Yokoly, Constant G. N. Gbalegba, David Malone, Benjamin G. Koudou.

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
