## [Decision Letter · Decision Letter 0]

3 May 2024

PONE-D-24-00912Assessing complex member composition and insecticide resistance status of Anopheles gambiae s.l. in three coastal health districts of Côte d'Ivoire.PLOS ONE

Dear Dr. Ives,

Thank you for submitting your manuscript to PLOS ONE. After careful consideration, we feel that it has merit but does not fully meet PLOS ONE’s publication criteria as it currently stands. Therefore, we invite you to submit a revised version of the manuscript that addresses the points raised during the review process.

You will see that one of the reviewers has raised several questions and clarification requests. Please make sure to address all of them indicidually.

We look forward to receiving your revised manuscript.

Kind regards,

Luca Nelli, PhD

Academic Editor

PLOS ONE

3. We note that your Data Availability Statement is currently as follows: [All relevant data are within the manuscript.]

5. Please include your tables as part of your main manuscript and remove the individual files. Please note that supplementary tables (should remain/ be uploaded) as separate ""supporting information"" files".

Reviewers' comments:

Reviewer's Responses to Questions

**Comments to the Author**

1. Is the manuscript technically sound, and do the data support the conclusions?

Reviewer #1: Yes

Reviewer #2: Partly

Reviewer #3: Yes

2. Has the statistical analysis been performed appropriately and rigorously? 

Reviewer #1: Yes

Reviewer #2: Yes

Reviewer #3: Yes

3. Have the authors made all data underlying the findings in their manuscript fully available?

Reviewer #1: No

Reviewer #2: Yes

Reviewer #3: Yes

4. Is the manuscript presented in an intelligible fashion and written in standard English?

Reviewer #1: No

Reviewer #2: Yes

Reviewer #3: Yes

5. Review Comments to the Author

Reviewer #1: Nice paper on Anopheles gambiae s.l. resistance level to pyrethroid insecticides and mechanisms involved in three coastal health districts of Côte d'Ivoire.

However, the level of English use lowered the quality of the manuscript. I would suggest the authors to really take time and rephrase most the sentences to an acceptable level.

Reviewer #2: To select the appropriate vector control products for us in a particular region of a country, it is important to determine which vector species are present and to establish the susceptibility of each species to the insecticides formulated into the products available.

The manuscript states that 'Dead and surviving mosquitoes’ exposure in WHO tube bioassays and CDC bottle tests were separately stored individually in Eppendorf tubes in silica gel and kept at −20 °C for further identification of An. gambiae s.l. complex members and kdr and Ace-1 mutations'. This would have facilitated the determination of the susceptibility of each molecular form of An. gambiae s.l. to each insecticide tested. However, the susceptibility data are presented simply as An gambiae s.l.. Assuming that the molecular form of each dead and alive mosquito in a susceptibility test has been determined as stated, it would be much more valuable to present the susceptibility data for each molecular form in each village.

There are numerous instances of the incorrect reference being identified in the text.

Reviewer #3: This manuscript describes the results of assessing complex member composition and insecticide resistance status of Anopheles gambiae s.l. in three coastal health districts of Côte d’Ivoire. The paper appears to be methodologically sound, and the analyses are well described in the manuscript. The strength of the paper is the fact that the authors are able to compare between three health districts,

Comments 1: Line 1 and 2. It’s will be important to reformulate the title of the manuscript

I’m suggested: Assessing of species composition and insecticide resistance of Anopheles gambiae complex member in three coastal health districts of Côte d’Ivoire

The aim of the study was to investigate the complex members and the insecticide resistance in the Anopheles gambiae s.l. populations in coastal Côte d'Ivoire.

a. Abstract:

You can choice to write in the manuscript between “An. gambiae s.L.” and “An. gambiae complex”. It's a totology when you write An. gambiae s.L. complex

Comment 1: Line 33: Author should put the 4.7% of hybrid species for what species?

Comment 2: Author should put the abbreviation (An. gambiae s.L) throughout the abstract subsections as the species name “Anopheles gambiae sensu lato” was already mentioned at the beginning of the abstract in line 41.

Comment 3: Abstract and throughout the manuscript, common Latin terms such as “s.s” and “s.L” are not italicized.

Comment 4: The authors are not mentioned in the abstract and the methodology if this is a cross-sectional survey, a longitudinal survey…etc. Add in the abstract and the methodology (Study site)

b. Introduction:

Ccomment 1: This is a study of the An. gambiae complex, except that I have only seen two species of the complex mentioned in the manuscript. The authors needs to define the An. gambiae complex and to cite the nine member of An. gambiae s. L. These articles will help you to cite the different species of the An. gambiae complex member, those which contribute to transmission and those which do not contribute. Loughlin SO (2020) https://doi.org/10.1080/20477724.2020.1722434. Abel et al. (2024) https://doi.org/10.1186/s13071-023-06102-7

Comment 2: The authors are not mentioned the Entomological Inoculation Rate (EIR) of An. gambiae s.L in coastal Côte d'Ivoire (in the studied sites).

c. Methodology:

Comment 1: Line 126 and 135. The larval method collected is not mentioned for what? What do you collected in the water wells, natural breeding sites, gardens field …etc larval only or not?

Comment 2: For rearing adult, author need to mention the technics used of rearing adult mosquito and the water type

Comment 3: Line 136: WHO tube bioassay assesses the degree of susceptibility for what?

Comment 4, Line 137 to 139: The author mentioned that the diagnostic dose tests of pyrethroids (alphacypermethrin, deltamethrin, and permethrin) were conducted in 2018, 2019, and 2020. The following tests were conducted exclusively in 2020 on the same population of An. gambiae complex (Fo, F1, F2…)?

Comment 5: The bottle analysis method is a monitoring tool for detecting and characterizing changes in insecticide susceptibility in vector populations?

Comment 6: Mosquito sample processing for DNA extraction was not mentioned clearly, does the author used whole mosquito for extraction or it was dissected to head and thorax and abdomen?

Identification of target site mutation (L202)

Comment 7: L202: For molecular Identification of target site mutation, author need to mention the primers used and PCR probes sequences and brief about the PCR methods.

Comment 8: L205-205: The author said that the krd (Kdr-East and Kdr-West) mutation was investigated using real-time PCR (TaqMan assay). To my knowledge, the probes (Kdr-East and West) attach to the FAM, I need to know how do you differentiate it? Did you do two separate the mix-PCR (WT, sensimix, water and Kdr-E FAM; WT, sensimix, water and Kdr-W FAM) or in the same mix did you put the Kdr-E and Kdr-W?

Data analysis

Comment 9: Please explain how this statistical analysis plan answer the stated question,: role of individual members of An. gambaie compex to malaria transmission?

Comment 10: Were there any considerations for possible non-blood-fed females from exophagic, zoophagic and endophilic mosquitoes? Any chance that some mosquitoes feed?

d. Results:

Species composition of the Anopheles gambiae complex

Comment 1, L236: For molecular identification of Anopheles gambiae complex member species, author mentioned the SINE200 or SINE-PCR using (Line 187 to 188). To my knowledge, SINE-PCR makes it possible to identify three species of the An. gambiae complex, namely: An. gambiae ss, An. arabesnsis and An. coluzzii. This is what I don't see in this manuscripts, in these results, the author identify two species and the hybride, were these others species in the complex member?

Comment 2: L237: the author said that 672 An. gambiae s.l. was selected from the three studied sites for species identification, however only 671 samples were used in An. gambaie s.l species molecular identification (there is 1 sample undetectable), because when I do the calculations of 593+51+27=671, where is the remaining 1 mosquito?

Comment 3: Figure 2 is not adopted for these results, remove that. It is better to represent the distribution of An. gambiae s.l. species on a table

Comment 4: L244, Remove the “of members” and write in the Table XX: Distribution of Anopheles gambiae s.l. in the three studied sites

Insecticide susceptibility in An. gambiae s.l

Comment 5: How many Anopheles have been testing with each insecticide, years and the different studied sites? The number of each An. gambiae s.L. alive and death (e.g: 2% (20/672))?

Intensity of resistance section

Comment 6, L273-273: The author said that: these results confirmed the strong resistance of An. gambiae s.L., for what? An. coluzzii or hybrid…etc.

Resistance mutation (L352)

Comment 7: Was the Kdr reading validated in relation to the value of the curve (CT), at the N point or by the value of FAM? What are the CT values (≤25 was positive in Kdr-E or W; 26-30 was RS-E or W) that you consider positive for the Kdr-E and the Kdr-W (RR)? For RS and SS?

- Wreting mistakes were recognized in line, L393 “Cô te d'Ivoire” and in, L394 “Bouaké ”.

Discussion

Comment 1: The authors should provide information, however, moderate resistance intensity was recorded with a 10X concentration of alphacypermethrin in the Aboisso populations, and additional investigations are required for better understanding. In this study, pre-exposure to PBO showed significantly increased pyrethroid-induced mortality in all An. gambiae s.l. populations. However, full susceptibility was not recovered using PBO, except with the strain of Jacqueville with deltamethrin. Is there any reports document to describe this resistance in Côte d’Ivoire?

Comment 2: L449-450. The author said that the pyrrole chlorfenapyr and neonicotinoid clothianidin have new modes of action and are good candidates for malaria vector control in areas comprised of mutation and/or metabolic resistance to pyrethroids. Can you say that you can advise the authorities of your country to change the molecule or the insecticide (pyrethroids) to impregnate the mosquito nets by pyrrole chlorfenapyr and neonicotinoid clothianidin in relation to this study? Explain

Comment 3: Please, author should revise his discussion accordingly from Line 407 to 460

Conclusions

Comment 1: Lines 463-463: Check whether the revised sentence: The present study showed that An. gambiae s.l. was mainly composed of An. coluzzii, followed by An. gambiae s.s and hybrids in costal Côte d’Ivoire

I suggested: The present study showed that An. gambiae s.l. was predominate by An. coluzzii, followed by An. gambiae s.s and hybrids in the three studied sites of costal Côte d’Ivoire

6. PLOS authors have the option to publish the peer review history of their article (what does this mean?). If published, this will include your full peer review and any attached files.

Reviewer #1: No

Reviewer #2: No

Reviewer #3: **Yes: **Jacques Dollon NTABI MBAMA

---

## [Author Response · Author response to Decision Letter 0]

11 Aug 2024

Reviewer 1

Comment Lines 36-37: suggest you say “induces …” instead of “restore”

Response : This has been done.

Comment Lines 45-46: Keywords: Anopheles gambiae s.l., pyrethroids resistance, PBO, Chlorfenapyr, clothianidin, Vgsc 995F, Vgsc 995S, Ace-1 280S Coastal Côte d'Ivoire

Response : This has been done. See lines 46 to 47 of manuscript.

Comment line 51:to be removed “some of”

Response :This has been done

Comment line 68 No need “burden”

Response :This has been done

Comment line 78 : Please consider revising these sentences and other as referring to the definition of species “Hybrids” individual cannot be considered as species.

Response : This has been done, see lines 81 to 83 of manuscript. “The species An. gambiae s.s., An. coluzzii, and the hybrids resulting from their interbreeding, all part of the An. gambiae complex, have been identified throughout the country”.

Comment line 89-92: Rephrase sentence from line 89 to 92 , as it may be confusing to reader. how susceptibility test can be carried out in order to have nets?

Response : The sentence was rephrased by : Insecticide sensitivity tests have also been conducted so that these data can facilitate the development of mosquito nets incorporating pyrethroids and other active ingredients (synergists, pyrroles, etc.) as well as combinations of active ingredients, for the upcoming LLIN distribution campaign, based on entomological stratification data. See lines 93 to 97 of manuscript.

Comment line 97: coma after “incidences” on line 97 and suggest change to “thousand” instead

Response : This has been done, see lines 104 to 105 of manuscript.

Comment line 110: I suggest you say “in Côte d’Ivoire” Instead

Response : This has been done, see line 117 of manuscript.

Comment line 120: should this be in present tense? In general, we use the present tense to describe actions and states of being that are still true in the present.

Response :This has been done, See line 126 of manuscript.

Comment line 130-131: There is a redundancy here with the prious sentence , considere rephrasing and/or combining them

Response : This has been done, see lines 136 to 137 of manuscript. The larvae and pupae were collected from larval habitats between January 2018 and December 2020.

Comment line 132: use “into” instead

Response : This has been done, see line 138 of manuscript.

Comment line 135: Species were identified morphologically using identification keys [26].

Response : The sentence has been revised and improved. See lines 141 to 142 of manuscript. Species (An. gambiae s.l.) were morphologically identified using identification keys [26] before being utilized for various sensitivity tests and molecular analyses.

Comment line 138-140: Consider combining the two sentences from lines 138 to 140, When were the tests done with the chlorfenapyr then?

Response : This has been done, see lines 146 to 148 of manuscript. The following tests, including intensity tests (1X, 5X, and 10X), synergist tests (PBO), and clothianidin tests, were exclusively conducted in 2020.

Comment line 143: “in each of the four testing tubes” instead

Response : This has been done, see lines 150 of manuscript.

Comment line 148: Please double check the PBO concentration on the coated paper, should this be 4% instead

Response : This has been rectified, see line 155 of manuscript.

Comment line 155-156 :There is need to moved thos down to line 160 before the sentence starting with “Mosquitoes…”

Response : This has been done, see lines 167 to 168 of manuscript.

Comment line 165 : Please consider revising the sentence on line 165 starting with “The tests were...” to make it clear to readers

Response : This has been done, see lines 173 to 174 of manuscript.

Comment line 166 : Is there any rational in testing against the diagnostic dose (100µg) and then 200µg (why not other concentration)

Response : Testing chlorfenapyr against An. gambiae at both the diagnostic dose (100µg) and a higher dose (200µg) serves multiple objectives. Firstly, it aids in establishing the effectiveness and potency of chlorfenapyr at the standard diagnostic concentration. Secondly, evaluating it at a higher concentration offers insights into its potential efficacy at elevated doses, particularly pertinent in scenarios of mosquito resistance or where higher concentrations are necessary for effective control.

Comment line 166-167 : I could not check the protocol in the given ref, so what guided the choice of these number insteaad of using 20 to 25 Mosquitoes. Why 2 days mosquitoes while it is recommend 3-5 old days moquitoes.

Response : The decision to use these specific numbers (15 to 20) instead of 20 to 25 mosquitoes was informed by Brogdon and Chan's 2012 protocol (https://stacks.cdc.gov/view/cdc/21777). Additionally, the age of the mosquitoes was determined based on a previous study conducted by Kouassi et al. in 2020 (DOI: 10.1186/s12936-020-03523-y).

Comment line 168 : Please check for the REF 29 on line 168, I am not sure if this is the good reference you should report here, as this refers to information related to epidemiology. Do add to this reference the link to make it easier to find it online.

Response: The accurate reference has been included, and the link is https://stacks.cdc.gov/view/cdc/21777.

Comment line 171-173 : rephrase the sentence from line 171, should be “...surviving moquitoes to exposure in WHO tubes….

Response: This has been done, see lines 179 to 180 of manuscript.

General comment: Please consider removing thoughout the manuscript, the “s.l.” or “complex members” as they indicates the same please revise all the citations in the manuscript as they do not aline with the reference listing.

Response: This has been done

Comment line 176 : Please check this, it does not seem to be correctly referenced, should it be [30]?

Response: This reference has been revised by [33] https://doi.org/10.4269/ajtmh.1987.37.37

Comment line 176-185: the text here seems to be copied from the manufacture guidelines/protocol. Please resentence all these and make it clear what you did instead OR to just refrered to it by citation

Response: The text has been rephrased, see lines 183 to 194 of the manuscript.

Comment line 188: please check for the right reference.

Response: This reference has been revised, see line 197 of the manuscript.

Comment line 200: please use “an” instead

Response: This has been done, see line 209 of the manuscript.

Comment line 208-209: Rephrase the sentence from lines 208 - 209, and use “master mix” instead of “mix” alone and do the same for the milar term troughout the manuscript

Response: This has been done, see line 226 of the manuscript.

Comment line 217: please check for the right reference [29].

Response: The reference has been updated by [37] : https://iris.who.int/handle/10665/250677

Comment: Is there a rational of using. I am not sure why you used ANOVA or xxx to compare proportion data here, as this a transformation before comparison. Instead I will recommend using the GLM a more modern and right way to analyse binomial data with higher power than analyses of transformed data. Since you did collection from different sites and different years I would suggest doing a GLM and assessing if MR varied between sites according to year of collection by using an interaction between site and year

Response: This has been done. See lines 240 to 243 and Table 3 of the manuscript. “The package ‘lsmeans’ version 2.30-0 was used for various analyses. The mortality recorded each year was compared between each insecticide (pyrethroid diagnostic concentration) and at the same site using the pairwise tests of the generalized linear model (GLM)”

Comment line 227 : Revising the sentence on line 227

Response: This has been done. See lines 247 to 248 of the manuscript.

Comment lines 232, 233 and 340: In the table 2 and the lines 232, 233 and 340, can you change the letters used in the formula to avoid any confusing for the readers as some of the letters (“SS”) you use there are also consider as genotypes (use for example “aa” and “AA”)

Response: This has been done. This has been done. See lines 252 to 253, and Table 5 of the manuscript.

Comment line 238: Please consider revising these sentences and other as referring to the definition of species “Hybrids” individual cannot be considered as species.

Response: This has been done. See lines 258 to 259 of the manuscript.

Comment line 240: On line 240 consider revising terms as An. gambiae ss and An. coluzzii are no longer consider as “forms” but “species”

Response: This has been done. See lines 260 to 261 of the manuscript.

Comment line 243: Revise the legend on the figure, we can really see the line on the pie charte indicating the hybrids proportions

Response: Figure has been replaced by Table 1

Comment line 272: Please be consistant in your writing, consider writing “Fig.” (example lines 247 ; 276) or “Figure” but do not mixte them

Response: This has been done, see line 286 of the manuscript.

Comment lines 274 - 276: the sentence is not very clearly explained, revise the english there

Response : This has been done, see lines 288 to 290 of the manuscript. In the resistance pyrethroid intensity tests, conducted on samples from Aboisso, Jacqueville, and San Pedro, mortalities remained below the 98% threshold when exposed to 1X and 5X concentrations of all insecticides.

Comment lines 276 - 279: Consider revising the english also the sentence from 276 to 279

Response : This has been done, see lines 290 to 293 of the manuscript. However, at a concentration of 10X, a high intensity of resistance was still observed across all three mosquito populations for the insecticides, except for a case of moderate intensity resistance recorded with alphacypermethrin 0.5% in the Aboisso populations, resulting in 100% mortality.

Comment Fig 4 : Could you indicate the number of mosquitoes exposed on the graph instead of the proportion as you did on fig 4

Response: This has been done, see fig 2 of the manuscript.

Comment line 281: Please consider using “from” instead of “of”

Response: This has been done, see lines 295 to 296 of the manuscript. The title has been revised by : Mortality rates of Anopheles gambiae s.l. population exposed to pyrethroid intensity tests from the three study populations.

Comment 295 : Please revise the title on line 295, Bioassays is a methods you cannot have a method within this table thus this cannot be a “bioassay” 

Response: This has been done, see lines 309 to 310 of the manuscript. “Mortality of An. gambiae s.l. to pyrethroids with and without piperonyl butoxide in the study population”

Comment lines 300 to 301: Sentence from lines 300 to 301, please be more specific here, bars are displaying mortality from different days

Response: This has been done, see lines 313 to 318 of the manuscript. “After seven days of observation, An. gambiae s.l. populations in Jacqueville and San Pedro were found to be completely susceptible to clothianidin. The San Pedro population achieved a 100% mortality rate by the sixth day. In Jacqueville, the sensitivity threshold was reached on the seventh day, with a mortality rate of 98.81%. However, in Aboisso, susceptibility was not fully restored after seven days of post-exposure observation, with the mortality rate only reaching 55.7%.”

Comment line 307: Please consider removing “of” on line 308

Response: This has been done, see line 321 of the manuscript

Comment line 314 : This is a figure showing results, how can it be a “susceptibility of An. gambiae s.l.”

Response: The legend has been corrected by “Mortality of Anopheles gambiae s.l. exposed to 2% clothianidin in 2020 from the three study populations.” See lines 328 to 329 of the manuscript

Comment line 318 : Please revise the title on line 318, “CDC bottle bioassays” is a methods you cannot have a method on this figure, thus this cannot be a “CDC bottle bioassay” 

Response: The legend has been corrected by “Mortality of Anopheles gambiae s.l. exposed to chlorfenapyr in CDC bottle bioassays in 2020 from the three study populations.” See lines 331 to 332 of the manuscript

Comment line 325 : please use “was” and remove “to” instead

Response: This has been done, see lines 336 to 337 of the manuscript.

Comment line Lines 328- 329: Please consider the following “The mutation allelic frequencies were 2.3% for An. gambiae s.s. in Aboisso population and 5.7% in San Pedro population” instead.

Response: This has been done, see lines 340 to 341 of the manuscript. “The mutation allelic frequencies were 2.3% for An. gambiae s.s. in Aboisso population and 5.7% in San Pedro population”.

Comment line 330 : Please consider “Hybird” instead. Sentence from lines 328 - 330: Since An. gambiae ss from Aboisso has this mutation , please removed An. coluzzii and Hybrid from the brackets so that it is clear to readres that you are referring to them

Response: This has been done, see lines 340 to 342 of the manuscript. “The mutation allelic frequencies were 2.3% for An. gambiae s.s. in Aboisso population and 5.7% in San Pedro population. For the population of Aboisso, and San Pedro mutation was not detected.”

Comment line 347-352: Please consider revising these sentences and other as referring to the definition of species “Hybrids” individual cannot be considered as species.

Response: This has been done, see lines 358 to 364 of the manuscript. “The current study showed that An. gambiae s.l. was composed of two sibling species, namely An. coluzzii, An. gambiae s.s., and the product of crossbreeding. Among the two sibling species, An. coluzzii was the predominant species (88.24%). The samples from Jacqueville and San Pedro were composed only of An. coluzzii. Both sibling species were represented in the Aboisso An. gambiae s.l. population. Among the sibling species, An. coluzzii (66.95%) was most abundant, followed by An. gambiae s.s. and the product of hybridization.”

Comment line 361-362: Please consider the tense, “...was...” instead of “...has been..”

Response: This has been done, see line 372 of the manuscript.

Comment line 364 - 365: Contrasting inference here, I do not think An. coluzzii is linked to urban but the type of breeding sites as they breed preferentially in permanent water bodies such us puddles and farming sites. In addition, I could not see this from the reference [36] you cite here

Response: This has been done, see lines 374 to 376 of the manuscript. “Anopheles coluzzii's preference for hosts and specific breeding habitats, especially its attraction to the littoral region, may account for its dominance in these zones.” The reference was revised by https://doi.org/10.1186/1472-6785-9-17.

Comment line 368: Please consider revising these sentences and other as referring to the definition of species “Hybrids” individual cannot be considered as species.

Response: This has been done.

Comment line 380: Please consider revising these sentences and other as referring to the definition of species “Hybrids” individual cannot be considered as species.

Response: This has been done, see line 378 of the manuscript.

Comment line 385: Please consider using “are” instead

Response: This has been done, see line 395 of the manuscript.

Comment line 403: Pesticide is a genral term of chemical that kills pests and insecticide is a types of pesticides used against insects. Thus, please, revise the following “…including pesticides and insecticides…”

Response: The sentence has been revised. See lines 413 to 415 of the manuscript.

Comment line 416: please revise the sentence, it is not clear enough to me.

Response: The sentence has been revised to “The kdr genes are known to confer resistance to pyrethroid insecticides, which are commonly used in long-lasting insecticidal nets (LLINs) and indoor residual spraying (IRS). If the kdr 995F and 995S mutations spread, the effectiveness of these malaria vector control tools (LLINs and IRS) would be compromised”. See lines 428 to 431 of the manuscript.

Comment line 446 : “restored”

Response: restored was replaced by achieved. See line 461 of the manuscript.

Comment line 453 and 456 : “restored”

Response: restored was replaced by achieved. See lines 468 and 471 of the manuscript.

Comment lines 465 - 466: please consider “

---

## [Decision Letter · Decision Letter 1]

16 Sep 2024

PONE-D-24-00912R1Assessing of species composition and insecticide resistance of Anopheles gambiae complex members in three coastal health districts of Côte d’IvoirePLOS ONE

Dear Dr. Ives,

Thank you for submitting your manuscript to PLOS ONE. After careful consideration, we feel that it has merit but does not fully meet PLOS ONE’s publication criteria as it currently stands. Therefore, we invite you to submit a revised version of the manuscript that addresses the points raised during the review process.

There is just one final minor comment from Reviewer #3 that needs to be addressed. Please make the requested change, or provide a justification if you believe it is unnecessary. Once this is done, I will be pleased to accept your manuscript for publication.

We look forward to receiving your revised manuscript.

Kind regards,

Luca Nelli, PhD

Academic Editor

PLOS ONE

Journal Requirements:

Additional Editor Comments:

Can you please adress this final comment from reviewer #3?

Add in abstract and methodology

Commnt 1: For the An. 32 gambiae s.l. populations were primarily composed of Anopheles coluzzii (88.24%, n = 312), followed by Anopheles gambiae sensu stricto (7.56%) and the crossbreed (4.17%), I am suggested that: In the total of 672 An. gambiae S.L. analysed, 27 (4.17%) mosquitoes were not discriminated against; thus, they are part of other species of An. gambiae complex

Comment 2: If the hybrids identified here do not belong to the An. gambiae complex and are not distinct species, why do you identify those in An. gambiae complex if you used PCR? I think your work is based on the member An. gambiae? Replace the hybrids with: other species of An. gambiae complex

Reviewers' comments:

Reviewer's Responses to Questions

**Comments to the Author**

1. If the authors have adequately addressed your comments raised in a previous round of review and you feel that this manuscript is now acceptable for publication, you may indicate that here to bypass the “Comments to the Author” section, enter your conflict of interest statement in the “Confidential to Editor” section, and submit your "Accept" recommendation.

Reviewer #2: All comments have been addressed

Reviewer #3: All comments have been addressed

2. Is the manuscript technically sound, and do the data support the conclusions?

Reviewer #2: Yes

Reviewer #3: Yes

3. Has the statistical analysis been performed appropriately and rigorously? 

Reviewer #2: Yes

Reviewer #3: Yes

4. Have the authors made all data underlying the findings in their manuscript fully available?

Reviewer #2: Yes

Reviewer #3: Yes

5. Is the manuscript presented in an intelligible fashion and written in standard English?

Reviewer #2: Yes

Reviewer #3: Yes

6. Review Comments to the Author

Reviewer #2: The authors have addressed well the comments and suggestions made on the original manuscript. They have provided a solid link between the members of the Anopheles gambiae s.l. species complex at the study sites and their insecticide resistance profiles, both in terms o insecticide susceptibility and the mechanisms of resistance they possess.

Reviewer #3: Add in abstract and methodology

Commnt 1: For the An. 32 gambiae s.l. populations were primarily composed of Anopheles coluzzii (88.24%, n = 312), followed by Anopheles gambiae sensu stricto (7.56%) and the crossbreed (4.17%), I am suggested that: In the total of 672 An. gambiae S.L. analysed, 27 (4.17%) mosquitoes were not discriminated against; thus, they are part of other species of An. gambiae complex

Comment 2: If the hybrids identified here do not belong to the An. gambiae complex and are not distinct species, why do you identify those in An. gambiae complex if you used PCR? I think your work is based on the member An. gambiae? Replace the hybrids with: other species of An. gambiae complex

Because this complex are nine species

7. PLOS authors have the option to publish the peer review history of their article (what does this mean?). If published, this will include your full peer review and any attached files.

Reviewer #2: No

Reviewer #3: **Yes: **Jacques Dollon NTABI MBAMA, PhD

Postdoc follow

---

## [Author Response · Author response to Decision Letter 1]

24 Sep 2024

Response 1: We thank the Reviewer #3 for drawing our attention on hybrids identified as member of An gambiae s.l. complex identified in our present study. Please, note that the 27 mosquitoes identified as hybrids are genuine hybrids and not undetermined species. Their identification was based on the Santolamazza et al. method, which has been validated in prior research. For instance, Fassinou et al. (2019, Trop Med Health, 11:47:23, DOI: 10.1186/ s41182-019-0151-z) explicitly described the use of the S200 X6.1 primer set to identify the hybrid form (M/S) between An. coluzzii and An. gambiae s.s., as indicated in the legend of Figure 2 (page 5) in their publication. Furthermore, Kacou et aL (2024, Med Vet Entomol, 2024; DOI: 10.1111/mve.12759), in their recent research conducted in Côte d’Ivoire, confirmed the identification of hybrids (M/S) using the same S200 X6.1 primer set method. Additionally, the identification of hybrids in Coast Côte d’Ivoire is novel discovery. Indeed, Caputo et al. (2020, Infect Genet Evol, 98:105215; DOI: 10.1016/j.meegid.2022.105215) identified the presence of An. coluzzii-An. gambiae s.s. hybrids in Coast and inland Anopheles gamabiae s.l. complex in Côte d’Ivoire. Therefore, based on these established approaches and previous findings in similar sites in Côte d’Ivoire, we confirm that the 27 mosquitoes identified in our study are indeed An. coluzzii-An. gambiae s.s hybrids within the An. gambiae complex.

Response 2: Thank you very much for your thoughtful comments and suggestions. Please as indicated above, we would like to clarify that all mosquitoes used in this study were morphologically identified as An. gambiae s.l. prior to molecular analysis. Moreover, to date, the only subspecies of the Anopheles gambiae complex identified in the district under study are An. coluzzii and An. gambiae s.s., along with hybrids of these two subspecies, as shown in previous studies by Tondossama et al. (2023; DOI: https://doi.org/10.3390/insects 14090758), Yokoly et al. (2021; DOI: https://doi.org/10.1186/s12936-021-03938-1) and Caputo et al. (2020; DOI: 10.1016/j. meegid.2022.105215). Therefore, the individuals encountered in our study could only be the result of crossbreeding between An. coluzzii (M) and An. gambiae s.s (S), making the term "hybrids (M/S)" the most appropriate. These hybrids are indeed part of the Anopheles gambiae complex, as confirmed by our PCR analysis. The two other Reviewers. We believe it is unnecessary to replace "hybrids" with "other species of the An. gambiae complex," as these individuals are hybrids from two subspecies (i.e. An. coluzzii (M) and An. gambiae s.s (S)) within the complex An. gambiae s.l. Therefore, no requested change has been made to our revised manuscript R1.

We hope this clarification addresses your concerns.

---

## [Editor Report · Decision Letter 2]

29 Sep 2024

Assessing of species composition and insecticide resistance of Anopheles gambiae complex members in three coastal health districts of Côte d’Ivoire

PONE-D-24-00912R2

Dear Dr. Ives,

We’re pleased to inform you that your manuscript has been judged scientifically suitable for publication and will be formally accepted for publication once it meets all outstanding technical requirements.

Kind regards,

Luca Nelli, PhD

Academic Editor

PLOS ONE
---

## [Editor Report · Acceptance letter]

7 Oct 2024

PONE-D-24-00912R2 

PLOS ONE

Dear Dr. Kouamé, 

I'm pleased to inform you that your manuscript has been deemed suitable for publication in PLOS ONE. Congratulations! Your manuscript is now being handed over to our production team.

Kind regards, 

on behalf of

Dr. Luca Nelli 

Academic Editor

PLOS ONE